# Annual Fluctuations in Winter Colony Losses of *Apis mellifera* L. Are Predicted by Honey Flow Dynamics of the Preceding Year

**DOI:** 10.3390/insects13090829

**Published:** 2022-09-12

**Authors:** Jes Johannesen, Saskia Wöhl, Stefen Berg, Christoph Otten

**Affiliations:** 1Fachzentrum Bienen und Imkerei, DLR-Westerwald-Osteifel, Im Bannen 38, 56727 Mayen, Germany; 2Institut für Bienenkunde und Imkerei, An der Steige 15, 97209 Veitshöchheim, Germany

**Keywords:** automated hive scales, autumn colony loss, foraging activity, growing degree days, honey flow period, TrachtNet, surveys, *Varroa destructor*, winter colony loss

## Abstract

**Simple Summary:**

Honey bee colonies are lost mostly during the winter, and loss rates (the proportion of dead colonies) may fluctuate highly between years. We investigated whether foraging activity—measured as the start of honey flow in spring and its magnitude in summer—influence loss rates in the following winter. Estimates of loss rates were gained from two surveys, in autumn and in winter, while foraging was investigated with automated hive scales during the foraging season in March–July. The surveys showed that high loss rates in autumn were followed by high loss rates in winter, and that high winter loss rates were followed by low loss rates in the following autumn. The fluctuations were influenced by the start of foraging in spring, where an early start in March resulted in high loss rates in autumn and winter, whereas a high intake of nectar in May–June led to lower loss rates. Together, the surveys and the foraging patterns suggest that colony loss rates in winter are influenced by the preceding one and a half years.

**Abstract:**

Winter loss rates of honey bee colonies may fluctuate highly between years in temperate climates. The present study combined survey data of autumn and winter loss rates in Germany (2012–2021) with estimates of honey flow—assessed with automated hive scales as the start of honey flow in spring and its magnitude in summer—with the aim of understanding annual fluctuations in loss rates. Autumn colony loss rates were positively and significantly correlated with winter loss rates, whereas winter loss rates were inversely related to loss rates in autumn of the following year. An early start of net honey flow in spring predicted high loss rates in both autumn and winter, whereas high cumulative honey flow led to lower loss rates. The start of net honey flow was related to temperature sums in March. Combined, the results implied that the winter loss rate in one year was influenced by the loss rate of the preceding winter and shaped by honey flow dynamics during the following year. Hence, the rate of colony loss in winter can be viewed as a cumulative death process affected by the preceding one and a half years.

## 1. Introduction

The western honey bee, *Apis mellifera* L., is one of the main pollinators of natural and commercial crops [1,2]. Honey production in the EU amounts to c. 218,000 tons/year [3], whereas the US honey industry has an estimated value of USD 300 million [4]. Pollination services and honey production are negatively affected by colony mortality, first of all, overwintering colony losses. Understanding the factors leading to colony losses are of concern for making management decisions and assessments of bee health.

Surveys have proven a reliable way of estimating overwintering losses in temperate climate zones [5,6,7,8,9,10,11], where loss rates may fluctuate highly between years. In Europe, loss rates may exceed 30% [6,7,12], and overwintering mortality rates in the US have been estimated at up to 53.3% [9,10]. COLOSS surveys of winter mortality rates also show that loss rates in Europe may vary considerably among geographical regions in any year [8]. In Germany, where most beekeeper operations are of similar size (96% of all registered operations have less than 26 colonies [13]) and have similar *Varroa destructor*-management strategies [14], annual winter loss rates have fluctuated between 6% and 30% since 2003 [15].

Given that beekeeping practices in most countries are more-or-less uniform, drivers of annual fluctuations of colony losses must be sought beyond structural beekeeping constraints. It is well-known that the physiology and phenology of the honey bees are governed by the abiotic environment [16,17,18]. Seeley and Visscher [16] concluded that “the timings of colony growth and reproduction are essential elements in the honeybee’s suite of adaptations for winter survival”. The start of brood development in honey bees in late winter is mainly driven by temperature but modulated by photoperiod [19]. Long and hot summers may reduce winter survival by reducing foraging resources that will constitute winter stores [20,21]. In a combined analysis of landscape, management, insect toxic load and weather variables, Calovi et al. [22] identified weather variables of the preceding year as overriding factors for predicting winter loss rates. In particular, precipitation and growing degree days, i.e., temperature sums, in the warmest quarter of the preceding year explained overwintering survival with 73.3% accuracy.

Although the above studies have found evidence for the importance of temperature for colony survival, a 20-year survey of *V. destructor* abundance in Central Europe found that raised temperatures in spring (March–May) and autumn (October) increase *V. destructor* incidences in autumn [23]. The study concluded that climatic effects affecting bee abundance and brood availability were the main drivers regulating *V. destructor* abundance. Infection rates with *V. destructor* mites and honey bee-associated viruses are reported to be the main factors influencing honey bee overwintering success [24,25,26], and management practices in relation to *V. destructor* control [9,27] are important for survival in winter.

Until now, there is little or no long-term European survey data of colony loss rates before winter (but see Steinhauer et al. [27] for USA) and how loss rates before the winter are related to overwintering success. If pre-winter loss rates predict winter loss rates and if factors influencing these losses can be traced in the foregoing months in a similar manner as winter losses to the preceding summer [20,21,22], it might create auto-correlative responses, which may help explain fluctuations in annual loss rates.

We studied this presumption by analysing the dependence between colony loss rates in late summer/autumn and the subsequent winter, and whether foraging activity (honey flow) in the months preceding late summer influenced these loss rates. The present study was based on 10-year (2012–2021) survey data of loss rates in (1) late summer/autumn and (2) winter, in combination with (3) estimates of honey flow dynamics obtained from automated, interconnected hive scales [28]. The honey flow variables were regressed against estimates of *V. destructor* incidences monitored in an independent study [15].

## 2. Methods

This study combines German national surveys of annual late summer/autumn (hereafter: autumn) and winter colony losses (2012–2021) conducted by the Centre for Bees and Apiculture in Mayen with estimates of regional honey flow patterns gained from automated hive scales through the German hive scale network “TrachtNet” [28]. We analysed the data at three geographic levels with the purpose of (i) evaluating repeatability of the results and (ii) for comparing whether few data points with many observations (higher level) or more data points but with less observations (lower level) resulted in similar results. The geographic levels were: (1) “Germany” (all data 2012–2021), (2) the federal state Rheinland-Pfalz (RP) (data subset of Germany) and (3) the RP administrative Regions “Trier” (TR), “Koblenz” (KO) and “Rheinhessen-Pfalz” (RHP) (2012–2021) and the Bavarian (BY) region “Unterfranken” (UF) (2014–2021), for which yearly hive scale data from 10+ scales were available. The former three regions make up RP; all regions are subsets of Germany. The levels of analysis correspond to the NUTS classification system 0 (Germany), 1 (federal states), 2 (administrative regions within states).

### 2.1. Surveys

Autumn loss rates were estimated with an on-line survey asking: (1) “how many colonies in summer and late summer did you prepare for the winter?” and (2) “how many of these colonies have already perished in late summer and autumn?” The surveys were online from approx. mid October until the end of November of each year (2012–2021). The online survey for estimating winter losses assessed mortality between December and April by asking the questions: (1) “how many colonies did you have before winter?” and (2) “how many of these (winter) colonies have you lost?” Winter losses were reported to COLOSS (see e.g., Gray et al. [8]) but here we included additional data that did not meet all requirements of COLOSS, e.g., we included apiaries without postal code information. The winter questionnaires were online from approx. 1.-30. April (2012–2021). Colony loss rates in both surveys were calculated as average rates within regions (total number of lost colonies/total number of managed colonies). Plausibility analysis of the data was done as described in Gray et al. [8]. In both surveys, the participants were asked to mention the country, federal state, administrative region and postal code in which the apiaries were located. Both surveys were anonymous.

The time of reporting a loss (answering the questionnaire) relative to the time of observing a loss differs between the questionnaires: autumn losses are generally reported at the time of observation, whereas winter losses are registered in spring, mostly without knowledge of when losses occurred in the preceding months. Hence, we temporally evaluated loss rates during the period of response of the autumn surveys (2013–2021). Due to different openings of the autumn questionnaire among years (5–7 weeks) and because the number of answers decrease in the last 2–3 weeks, we grouped the answers into four standard weeks. In this analysis, 2012 was omitted due to a prolonged answering period, making a weekly evaluation difficult. Yearly average autumn colony losses were regressed against the response variable yearly average winter loss for samples N > 20 answers.

### 2.2. Automated Hive Scale Network “TrachtNet” and Variables

The TrachtNet network was established in 2011. Automated TrachtNet hive scales measure weights in 5 min intervals with a resolution of 5 g. Hive scales measure absolute and cumulative weights. The cumulative weight is the sum of weight gains (+) and losses (−). Weight changes exceeding 200 g within the 5 min interval are not considered in calculations of cumulative weight, thus, beekeeper-related actions are eliminated in the cumulative estimate. The corrected cumulative weight is therefore a measure of bee activity and forage intake. TrachtNet connects hive scales into so-called “virtual scales”, which estimate regional and cumulative foraging patterns (https://dlr-web-daten1.aspdienste.de/cgi-bin/tdsa/tdsa_client.pl) (accessed on 8 September 2022).

The cumulative weight curve of each year is calculated by setting the cumulative weight of a hive to zero on 1 January. A “standard” cumulative weight curve decreases through January and February, reaching a minimum cumulative weight (MinCW) in March or April. This is followed by a net gain when the forage input of nectar, pollen and/or honeydew exceeds consumption. The curve typically reaches a maximum in July (Figure 1a). MinCW may exhibit “local minima” in March and April (Figure 1b). Such minima occur when periods of net gain are followed by cold or rainy weather periods with net loss.

In this study, we distinguished between absolute MinCW (aMinCW) and first MinCW (fMinCW) (Figure 1b). The start of honey flow was defined as the calendar day of MinCW (aMinCal or fMinCal), i.e., the start of net weight gain, whereas the end of honey flow was defined as the maximum cumulative weight (MaxCW) and its calendar day (MaxCal). The minimum and maximum cumulative weights were calculated as deviations from the zero baseline, giving a cumulative weight range of honey flow RangeCW = MaxCW − MinMW. The honey flow duration in days was HF-Period = MaxCal − MinCal.

We considered three of these variables: fMinCal, RangeCW and HF-Period. fMinCal and aMInCal were identical in 8 out of the 10 years in Germany, 9 of 10 in RP and 32 of 38 in the Regions, hence correlated, where fMinWC had the best fit with loss rates (see below). RangeCW is a summary variable of MaxCW and MinCW, and HF-Period describes MaxCal (based on MinCal). The variable MinCW is principally an independent variable but it defines MinCal, which is the point of interest of the present study (and it is included in RangeCW) and was omitted from further analysis. The average number of hive scales sending data between March and July ranged from 97 in 2012 to 467 in 2021.

### 2.3. Temperature Estimation

Estimates of temperature were obtained from two sources. First, we calculated cumulative temperature sums (growing degree days, GDD) for RP and the four administrative regions. GDD was calculated with daily arithmetic mean temperatures (T_am_) (based on hourly means) as GDD = ∑(T_am_ − T_base_), where T_base_ = 5 °C, with a cut off at maximum temperature T_max_ = 30 °C. T_am_ = 0 if T_am_ < T_base_, if T_am_ > T_max_ then T_am_ = T_max_. GDD was related to the calendar day of MinCW (GDD-Day) and mean monthly GDD (January–May). The number of weather stations used for calculating T_am_ in RP was 140 (2012) – 180 (2021); in Unterfranken 42 (2014) – 52 (2021). T_am_ was calculated using the online platforms of the Agricultural Meteorology of RP (https://www.wetter.rlp.de/Agrarmeteorologie) (accessed on 8 September 2022) and BY (https://www.wetter-by.de/Agrarmeteorologie-BY/Landwirtschaft/Ackerbau/Temperatursummen-Mais) (accessed on 8 September 2022). Estimates of GDD were not available for the whole of Germany. To assess interactions with temperature here, we used mean monthly temperatures (2 m above ground) provided by the German Weather Service DWD (https://opendata.dwd.de/climate_environment/CDC/regional_averages_DE/monthly/air_temperature_mean/) (accessed on 8 September 2022).

### 2.4. Correlations with Varroa Destructor Abundance

We correlated TrachtNet variables with estimates of *V. destructor* infestations at the level of Germany. The infestation levels were estimated by the German bee monitoring programme [15] as the number of *Varroa*/100 bees/colony. The infestation estimates were based on 1000–1200 colonies sampled annually in summer (July) and autumn (October), respectively, in this programme.

### 2.5. Statistics

All statistical analyses were performed with JMP v.13.2.1, SAS Institute Inc.: Cary, NC, USA [29]. We first searched for linear associations between yearly autumn and winter loss rates and the continuous variables fMinCal, RangeCW, HF-Period and GDD-Day using the function “multivariate”. Thereafter, we applied multiple regression analysis with backward selection to assess the most important explanatory variables for autumn and winter mortality rates. The Regions data was analysed for interactions to degree. The relationship between loss rates and average monthly temperature (Germany and RP), temperature sums (RP and Regions) and *V. destructor* counts (Germany) were done in independent regression analyses.

## 3. Results

### 3.1. Loss Rates in Autumn and Winter

Reported colony loss rates in autumn (0.022–0.059) were significantly and positively associated with loss rates of the following winter (0.086–0.208) at all three levels of investigation: (Germany: *r*^2^ = 0.703, F_1,8_ = 18.99, *p* = 0.0024; RP: *r*^2^ = 0.763, F_1,8_ = 25.77, *p* = 0.001; Regions: *r*^2^ = 0.658, F_1,36_ = 69.269, *p* < 0.001). Winter colony loss rates was significantly inverse related to colony loss rate in the following autumn in Germany (*r*^2^ = 0.463, F_1,8_ = 6.034, *p* = 0.044) and Regions (*r*^2^ = 0.236, F_1,33_ = 9.914, *p* = 0.004)), with a similar trend but marginally significant in RP (*r*^2^ = 0.351, F_1,8_ = 4.531, *p* = 0.071) (Figure 2). Colony loss rates in autumn became higher with the ongoing calendar week of answering the questionnaire (ANOVA F_4,40_ = 36.293, *p* < 0.001) with significant effects between week 4 and lower (Figure 3). The actual losses per calendar week are reported in Appendix A.

### 3.2. Honey Flow Variables and Loss Rates

Multivariate analysis identified significant correlations between fMinCal and colony losses in autumn and winter in Germany (*p* = 0.0044 and *p* = 0.0160, respectively) (Figure 4a) and among Regions (*p* = 0.0029, *p* = 0.0026) (Figure 4c), where an early start of honey flow resulted in higher losses in the following winter. An identical trend was observed in RP but was not significant (*p* = 0.0695, *p* = 0.1358) (Figure 4b). There were no significant correlations with calendar day of absolute minimums (aMinCal) in Germany and Regions. The cumulative weight range (RangeCW) was correlated with loss rates in both autumn and winter: Germany (*p* = 0.0218, *p* = 0.0039), RP (*p* = 0.0206, *p* = 0.0075) and Regions (*p* = 0.0049, *p* = 0.0002) (Figure 4d,e), where high forage input was related to low mortality rates. The two variables fMinCal and RangeCW were independent in RP and Regions (*p* > 0.15) but marginally correlated for Germany (*p* = 0.05). The period of honey flow (HF-Period) and GDD of fMinCal (GDD-Day) were not associated with loss rates. Summary results are shown in Table 1. Honey flow data for Germany, RP and Regions are reported together with loss data (see above) in Appendix A.

Multiple regression analysis with backward selection showed that the explanatory variables for autumn loss rates was reduced to fMinCal (*p* = 0.0044, as above) in Germany and to RangeCW in RP (*p* = 0.0206, as above), whereas Regions was explained by both fMinCal (*p* = 0.0090) and RangeCW (*p* = 0.0156). Explanatory variables for winter loss rates were reduced to RangeCW in Germany (*p* = 0.0039, as above) and in RP (*p* = 0.0075, as above) whereas winter loss rates in Regions were influenced by fMinCal (*p* = 0.0014) and RangeCW (0.00262) and the interaction fMinCal*HF-Period (*p* = 0.0225).

In a second multiple regression analysis, we included autumn loss rates with the variables fMinCW and RangeCW to test their combined effect on winter loss rates. After backward selection, autumn loss rate remained the only explanatory variable in Germany and RP (*p*-values as above) whereas autumn loss rate (*p* < 0.0001) and RangeCW (*p* = 0.0127) explained winter loss rates in Regions.

### 3.3. Honey Flow and Temperature

In the Germany data set, the average temperature of March was associated with the calendar day of first MinCW (fMinCal) (R^2^ = 0.612, *p* = 0.007). Mean temperatures in February and April were not (*p* > 0.30). In RP, fMinCal was associated with mean temperature in March (R^2^ = 0.624, *p* = 0.007) and GDD in March (R^2^ = 0.736, *p* = 0.002) and April (R^2^ = 0.673, *p* = 0.004). However, because GDD is a cumulative value (mean temperature is not), GDD for April is an autocorrelation with GDD in March. RangeCW was not related to mean temperatures or to GDD. The temperature data are reported in Appendix A.

The calendar day of the beginning of net forage intake (fMinCal) was not associated with the temperature sum on that day (GDD-Day) (*p* > 0.70, only RP and Regions). Post hoc inspection of the data from the three adjacent administrative regions in RP revealed that yearly fMinCal was not significantly different between any of the three regions (paired *t*-tests, all *p* > 0.30), whereas yearly temperature sums on GDD-Day differed significantly between KO/TR (*p* = 0.004) and KO/RHP (*p* = 0.002), and was marginally significantly different between TR/RHP (*p* = 0.06) (GDD-Day 10 yr means: RHP *=* 82.5 > TR = 73.3 > KO = 67.5) (Figure 5). Hence, the calendar day of honey flow beginning was “synchronised” among the three adjacent regions and independent of significantly different temperature sums on the day of honey flow beginning.

### 3.4. Honey Flow Variables and Varroa Destructor

We related the honey flow variables fMinCal and RangeCW to *V. destructor* infestation levels reported in an independent study [15] (Figure 6). fMinCal was correlated with *V. destructor* infestation in July (R^2^ = 0.629, *p* = 0.006) and marginally significant with *V. destructor* infestation in October (R^2^ = 0.412, *p* = 0.045), where an early begin of net forage intake was correlated with higher rates of *V. destructor* later in the year. RangeCW was significantly inverse related to *V. destructor* infestation in October (R^2^ = 0.615, *p* = 0.007) and marginally significant in July (R^2^ = 0.418, *p* = 0.043). Low forage intake predicted higher *V. destructor* infestation rates.

## 4. Discussion

In this study, we combined estimates of honey flow dynamics obtained from interconnected, automated hive scales with autumn and winter colony loss rates to investigate annual fluctuations in winter loss rates and associations with loss rates before and after winter. Two honey flow variables, the calendar day of first minimum cumulative weight (fMinCal) and the range of cumulative weight (RangeCW), both predicted both autumn and winter losses. An early onset of net honey flow—and temperature sums in March—were related to higher loss rates in autumn and the following winter. By contrast, high cumulative weight gains during the foraging season, i.e., good foraging conditions, predicted low mortality rates, particularly those through the winter. The two variables were, except for one marginally significant value, independent.

Autumn loss rate was the strongest predictor of winter loss rate, and annual autumn colony loss rates increased as a function of the calendar week of response to the autumn survey. In other words, fMinCal and RangeCW influenced losses leading up to the subsequent winter. The survey data further indicated that autumn loss rates were inversely associated with loss rates of the preceding winter. The inverse association contradicts findings from a two-year study by Jacques et al. [12] who reported a positive relationship throughout Europe, but it is unclear if the authors refer to individual beekeeper operations or to average annual loss rates. The negative correlation derives from the amplitude of the loss rates in winter and in the subsequent autumn. In natural populations, temporal autoregressive components generally run over several years, e.g., [30,31], not between consecutive years as in the present study. We hypothesise that fluctuations and the amplitude in annual winter-colony loss-rates are related to bee health, which is modulated by honey flow dynamics via the climate. As a thought experiment, susceptible colonies might be “purged” in winters following years with honey flow conditions favouring high loss rates. This will leave “strong” colonies surviving the winter and lead to lower general loss rates in the subsequent autumn and winter, though these rates will be influenced by honey flow conditions during that year. It suggests that overwintering losses in any one year is the end result of a cumulative death-process influenced by the preceding one and a half year.

Previous studies have shown that temperature (sums) in summer are related to winter mortality rates [20,22]. Our results revealed that winter mortality rates in a temperate climate were influenced by temperature and bee activity already in the spring of the preceding year. Although these results were validated at different geographic levels, the data also indicated that other abiotic variables must interact with loss rates. First, the calendar day of honey flow begin (fMinCal) in adjacent regions in RP was not related to the temperature sum on that day (GDD-Day). We also did not find an association between RangeCW and temperature (sums) in April–June during the period of net honey flow (results not shown), and RangeCW was not related to the period of net weight gain. The period usually ends around 1 July, being more-or-less constant among the years. To better understand how RangeCW modulates loss rates—in particular winter loss rates—we need more information about abiotic factors affecting colony development during the honey flow period. Variables such as precipitation and humidity [18,20,22,32,33,34], temperature related flight activity [34,35,36] and availability of forage resources [37] may co-influence colony development. Although Clovani et al. [22] did not find effects of forage quality and abundance (landscape factors) on overwintering success, our study found a positive effect of forage intake.

Our study had two possible biases. First, loss rates in the winter surveys were probably inflated by some (anonymous) respondents including autumn losses in their answers of winter losses. This will affect the degree of the autumn-winter association, but it should not affect the dependency itself. By contrast, the reported autumn loss rates were not biased by winter loss rates, thus, the inverse winter-autumn association was unbiased. Another potential error at the Germany level of investigation was the geographic distribution of hive scales and the number of responses to the surveys. Both were skewed towards the west and south of Germany. Whereas the latter characterises the actual distribution of beekeepers and colonies in Germany, being more numerous in the west and south [13], the former is historically related to the development of TrachtNet. Because the density distributions of the two were correlated (except for the south-western state of Baden-Württemberg), the bee activity pattern for Germany more specifically supported interactions relative to the *density* of beekeepers and colonies. However, this geographic bias did not exist for RP or for the four administrative regions. Similar interactions between honey flow variables, March temperature and loss rates here and at the Germany level indicate that Germany results were robust.

In our correlative study, annual average *V. destructor* infestation levels in July and October estimated by an independent monitoring study [15] were significantly associated with fMinCW (together with March temperature) and RangeCW (Figure 6). The two variables fMinCal and RangeCW had opposite and temporally different associations with *V. destructor*; an early fMinCal predicted high *V. destructor* counts in July whereas high RangeCW was associated with low *V. destructor* counts in October. The results therefore not only indicated a time-related influence of the two variables on colony loss rates but also different time-related interactions with *V. destructor*. We note here that the associations with *V. destructor* need confirmation at other levels of analysis. The results support studies identifying temperature to be a main factor influencing *V. destructor* development and subsequently winter losses [23], and that longer brood periods are involved in raised levels of *V. destructor* infestation [38]. By contrast, good foraging conditions may strengthen bee health, promote reproduction and increase the number of young bees, and reduce susceptibility to infections, also in years with long brood periods. In particular, pollen promotes health against a variety of parasites and pathogens [39,40]. Pollen flow typically continues after the maximum cumulative weight is reached.

The present study tested whether autumn and winter colony loss rates were related and whether bee activity, measured as honey flow activity, in spring and summer influenced colony survival rates. Affirmative evidence was found for both. Simulation studies analysing winter losses based on summer weather conditions have found good predictive accuracy [22]. Including spring weather and honey flow variables but also loss rates of the preceding year will likely improve the predictive value of such models. In particular, the positive effect of cumulative weight gain needs more consideration. Automated hive scales connected together to form regional virtual scales as in TrachtNet offer the possibility to address these questions further.

## 5. Conclusions

The present study showed that colony loss rates in both autumn and winter were linked to honey flow dynamics of the preceding flowering season, and that high loss rates in winter were associated with low autumn loss rates in the following year. The results also indicated that an early start of honey flow as well as a poor foraging season may benefit *V. destructor* development. The between-year association of loss rates implied that loss rates in one winter were influenced by the loss rate in the previous winter, but will be modulated by honey flow during the year. This dependency may cause auto-correlations between years, which may help explain the amplitude of annual loss rates. The implications of the study for the apicultural industry are that management measures can be adjusted to ambient conditions of the abiotic environment.

## Figures and Tables

**Figure 1 insects-13-00829-f001:**
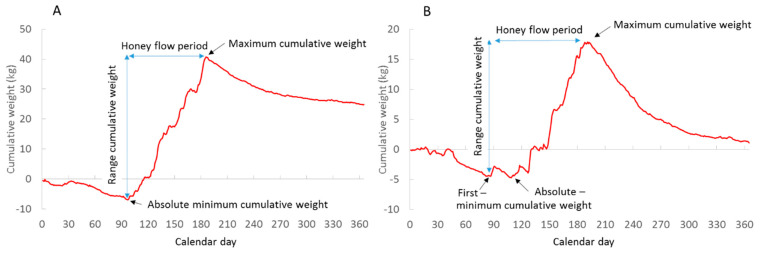
Cumulative forage curves showing a standard curve (**A**) with a single absolute minimum cumulative weight (Germany 2015) and (**B**) a curve with a first minimum that differs from the absolute minimum (Germany 2021).

**Figure 2 insects-13-00829-f002:**
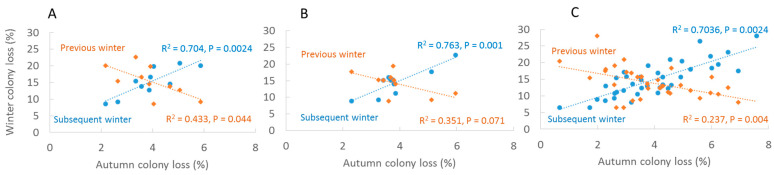
Associations between autumn loss rates and loss rates of the previous and subsequent winter in Germany, 2012–2021. (**A**) Annual averages Germany, (**B**) annual averages RP, (**C**) annual averages administrative regions.

**Figure 3 insects-13-00829-f003:**
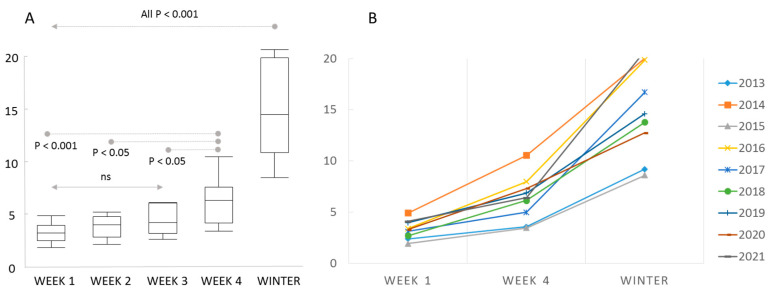
Colony loss rates relative to the standardised week of response to the late summer/autumn survey. The winter colony loss rates are shown for comparison. (**A**) Box plots of averages among years with significance estimates. (**B**) Annual loss rates in autumn in response week 1 and week 4 and winter loss rates. The loss rates increase between week 1 and week 4 each year.

**Figure 4 insects-13-00829-f004:**
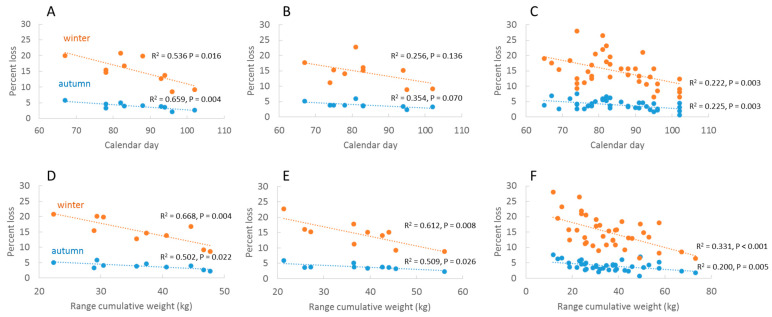
Annual colony loss rates in autumn and winter as a function of calendar day of first minimum cumulative weight (fMinCal) and range of cumulative weight (RangeCW) at three geographic levels of analysis: Germany (**A**,**D**); RP (**B**,**E**); administrative regions (**C**,**F**).

**Figure 5 insects-13-00829-f005:**
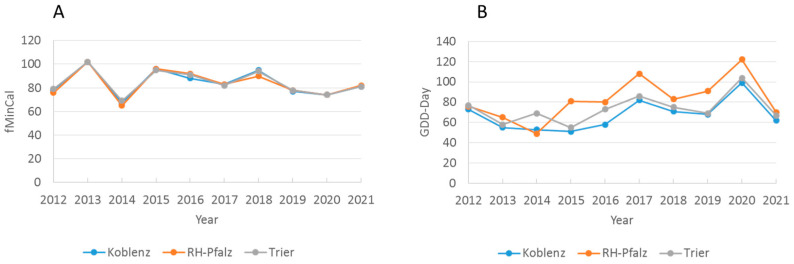
Start of honey flow and temperature sums of net cumulative honey flow in adjacent RP-regions, Koblenz, Rheinhessen-Pfalz (RH-Pfalz) and Trier 2012–2021. (**A**) Calendar day of first minimum cumulative weight (fMinCal), and (**B**) temperature sum on that day (GDD-day). The calendar day was constant among regions per year, whereas GDD-day differed significantly between regions (see text).

**Figure 6 insects-13-00829-f006:**
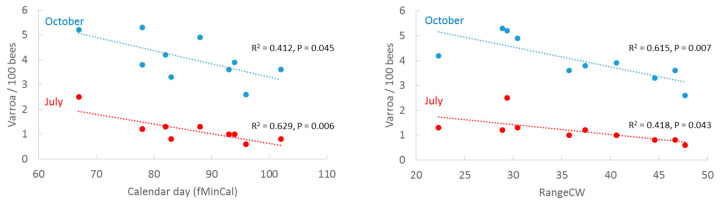
Annual average infestation levels of *V. destructor* per 100 bees in Germany (2012–2021) in July and October [15] as a function of calendar day of first minimum cumulative weight (fMinCal) and range of cumulative weight (RangeCW).

**Table 1 insects-13-00829-t001:** Correlation coefficients and significance levels of honey flow variables as predictors of autumn and winter colony loss rates. N = number of data points, GDD = growing degree days. For an explanation of geographic levels (Germany, RP, Regions), please refer to text. The number of observations are reported in Appendix A.

		Autumn Loss			Winter Loss		
Variable	Variable Abbr.	Germany (N = 10)	RP (N = 10)	Regions (N = 38)	Germany (N = 10)	RP (N = 10)	Regions (N = 38)
Calendar day, first minimum	fMinCal	–0.812 **	–0.595 +	–0.471 **	–0.732 *	–0.506 ns	–0.475 **
Calendar day, absolute minimum	aMinCal	–0.535 ns	–0.602 +	–0.246 ns	–0.260 ns	-0.464 ns	–0.298 +
Honey flow period	HF-Period	0.562 ns	0.378 ns	0.208 ns	0.501 ns	0.014 ns	0.080 ns
Honey flow range	RangeCW	–0.709 *	–0.713 *	–0.447 **	–0.817 **	–0.783 **	–0.576 ***
GDD on calendar day	GDD-Day	-	–0.129 ns	–0.207 ns	-	–0.186 ns	–0191 ns

- not applicable, ns = not significant, + < 0.10, * < 0.05, ** < 0.01, *** < 0.001.

## Data Availability

The data for this study is available in the Appendix A.

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
