# Peer review of "Annual Fluctuations in Winter Colony Losses of Apis mellifera L. Are Predicted by Honey Flow Dynamics of the Preceding Year"

_insects, 2022, doi:10.3390/insects13090829_

Round 1
Reviewer 1 Report
Dear Authors,
the paper is of great interest for the scientific and beekeeping community as the use of IoT tools is more and more common and a "predictive" use of such data could give many benefits to honey bee health and improve beekeeping management practices. Finally, making huge data collection of interconnected hive scales "meaningful" is fundamental. I added only few comments to improve readability.

Author Response
Dear Authors, the paper is of great interest for the scientific and beekeeping community as the use of IoT tools is more and more common and a "predictive" use of such data could give many benefits to honey bee health and improve beekeeping management practices. Finally, making huge data collection of interconnected hive scales "meaningful" is fundamental. I added only few comments to improve readability.
59. Amended as suggested
73 -79. We have rewritten the section for more clarity.
98 -119. Amended as suggested
124. The accuracy has been specified (5g) – thank you for mentioning this. We have not specified “bee activity” in more detail but have restructured the paragraph. In the current paper, bee activity is hive-weight increment (+ or -) resulting from honey flow. But the 5-min / 5g resolution actually allows estimates of when bees (and the amount thereof) leaving and re-entering the hive. During the night, we can trace water evaporation (nectar converted to honey).
Fig. 1. Fonts have been made bigger
167-170. The paragraph has been rewritten. We still do not specify the annual numbers of colonies studied by DeBiMo. The numbers are listed in the reports of this programme.
Fig. 2 Font size increased in original figure
311. Corrected
Reviewer 2 Report
It is a pleasure for me to review your work concerning the honeybee loss in Europe, it is very interesting work.
I have only very few comments that could help in improving the final published version of the manuscript:
- The title should be improved the present one is not attractive and not easy to get the idea about the work very well from the present title.
- Many parts of the material and methods show ambiguity in meaning. the way and style of writing of this section need to be improved.
- you need to give a conclusion of your work and how such kind of work will impact the Apiculture industry.
Author Response
It is a pleasure for me to review your work concerning the honeybee loss in Europe, it is very interesting work.
I have only very few comments that could help in improving the final published version of the manuscript:
- The title should be improved the present one is not attractive and not easy to get the idea about the work very well from the present title.
We have changed the title, hopefully making it more attractive.
- Many parts of the material and methods show ambiguity in meaning. the way and style of writing of this section need to be improved.
We have revised the section, keeping (mostly) to the past tense. The TrachtNet paragraph has been amended. It now describes the network in more detail (but not extensively) and we have corrected the tenses. In the TrachtNet paragraph we still mix the tenses: “what we did” is written in the past tense while “what TrachtNet does” is written in the present tense. We hope there is less ambiguity.
- you need to give a conclusion of your work and how such kind of work will impact the Apiculture industry.
A short conclusion with suggestions for applied beekeeping are now included
Reviewer 3 Report
In this study, the authors present some important findings. However, I have some comments that should be addressed to improve the quality of this article.
Simple summary
· Line 11: Please replace the proportion of colonies that die with the proportion of dead colonies.
Abstract:
· Should include background, objective, materials and methods, results, and conclusion. Please include the missed.
· Line 25, 26: foraging activity (honey flow), foraging activity not mean honey flow.
· Line 29: Please replace low by the flow.
Introduction:
· Line 40: The author can use the following reference: Al-Kahtani, S.N., Taha, E-K.A.; Al-Abdulsalam, M. (2017). Alfalfa (Medicago sativa L.) seed yield in relation to phosphorus fertilization and honey bee pollination. Saudi Journal of Biological Science, 24(5): 1051-1055.
· Line 60: The author can use the following reference: Taha, E-K.A.; Al-Kahtani, S.N. (2019). Comparison of the activity and productivity of Carniolan (Apis mellifera carnica Pollmann) and Yemeni (Apis mellifera jemenitica Ruttner) subspecies under environmental conditions of the Al-Ahsa oasis of eastern Saudi Arabia. Saudi Journal of Biological Science, 26(4): 681-687.
· Line 69: Please replace Infection with Infestation.
· Line 73: Please replace see [27] by Steinhauer et al. [27].
· The aim of the study needs to write at the end of the Introduction.
Material and methods
· Line 104: Please replace (see e.g. [8]) with (see e.g. Gray et al. [8]).
· Line 108: Please replace described in [8] with described by Gray et al. [8].
· Line 265: Please replace Infection with Infestation.
· Statistical analysis: the design of the experiment should be mentioned.
Results
· In Figs. 2 and 4: delete the words blue and orange from fig. capture and select show legend from chart elements.
· In Fig. 6: delete the words blue and red from fig. capture and select show legend from chart elements.
Discussion
· The discussion needs improvement.
· Line 332: Please replace By contrast, with On contrast.
· Line 334: Please replace Infection with Infestation.
· Lines 337 and 338: the aim should be at the end of the Introduction.
· Line 475: Please replace Infection with Infestation.
· The authors have used honeybee and honey bee, please use one of them I suggest honey bee throughout the manuscript.
Author Response
Simple summary
Line 11: Please replace the proportion of colonies that die with the proportion of dead colonies.
done
Abstract:
Should include background, objective, materials and methods, results, and conclusion. Please include the missed.
The Abstract has been amended accordingly. Background: fluctuations, MM: surveys and hive scales, results: most important dependences, conclusion: fluctuations result from year-through death process.
Line 25, 26: foraging activity (honey flow), foraging activity not mean honey flow.
We have amended the wording and define foraging activity in the Abstract as: foraging activity – measured as the start of honey flow in spring and its magnitude during the summer. We are aware that forage resources include other resources than honey; pollen for example. Hive scales measures the total weight changes. Because many, if not most, beekeepers relate foraging to honey flow, we used the two more-or-less as synonyms.
Line 29: Please replace low by the flow.
done
Introduction:
Line 40: The author can use the following reference: Al-Kahtani, S.N., Taha, E-K.A.; Al-Abdulsalam, M. (2017). Alfalfa (Medicago sativa L.) seed yield in relation to phosphorus fertilization and honey bee pollination. Saudi Journal of Biological Science, 24(5): 1051-1055.
Line 60: The author can use the following reference: Taha, E-K.A.; Al-Kahtani, S.N. (2019). Comparison of the activity and productivity of Carniolan (Apis mellifera carnica Pollmann) and Yemeni (Apis mellifera jemenitica Ruttner) subspecies under environmental conditions of the Al-Ahsa oasis of eastern Saudi Arabia. Saudi Journal of Biological Science, 26(4): 681-687.
We have chosen to not cite the papers. Regarding the first paper, we referred to the generality of pollination rather than specific plants (and the efficiency of honey bee pollination in alfalfa – at least in northern parts of the world – is debated). Regarding the second paper, it was actually considered in an early version but it didn’t quite fit into the paper later.
Please excuse us for not citing the papers.
Line 69: Please replace Infection with Infestation.
Done. We have changed the wording throughout the paper.
Line 73: Please replace see [27] by Steinhauer et al. [27].
done
The aim of the study needs to write at the end of the Introduction.
We have rewritten the last paragraph of the introduction to emphasise the aim. However, we do not explicitly write “aim” but rather “testing the presumption” (which is derived in the previous sentence). We now include the aim in the Abstract.
Material and methods
Line 104: Please replace (see e.g. [8]) with (see e.g. Gray et al. [8]).
done
Line 108: Please replace described in [8] with described by Gray et al. [8].
done
Line 265: Please replace Infection with Infestation.
done
Statistical analysis: the design of the experiment should be mentioned.
The design is mentioned in the methods. It is a multivariate correlative study. It is not an experimental study.
Results
In Figs. 2 and 4: delete the words blue and orange from fig. capture and select show legend from chart elements.
Because of the composite nature of the figures (multiple graphs, each with dual regressions), we chose to present the figures (and text) in this manner. We believe, in this case, that this type of presentation is easier for the readers to interpret than chart elements for each graph will be. We have therefore refrained from changing the text.
In Fig. 6: delete the words blue and red from fig. capture and select show legend from chart elements.
Please refer to reply to Fig. 2 and 4.
Discussion
The discussion needs improvement.
We have made a major revision of the discussion
Line 332: Please replace By contrast, with On contrast.
Not changed. “By contrast” is – we believe – a standard expression
Line 334: Please replace Infection with Infestation.
done
Lines 337 and 338: the aim should be at the end of the Introduction.
Please refer to response concerning the aim above
Line 475: Please replace Infection with Infestation.
done
The authors have used honeybee and honey bee, please use one of them I suggest honey bee throughout the manuscript.
We changed to the word to honey bee (own text, not references), as suggested
Reviewer 4 Report
Dear Editor,
The authors of this manuscript evaluated the influence of foraging activity on yearly fluctuations in honey bee losses in Germany.
I have some suggestions to improve the paper.
Line 28. high cumulative honey flow – please, give some details of this statement.
Line 30-31. And winter losses were inversely related to autumn losses in the following year - I see no logic in this statement. Please, explain in more detals.
Line 69-70. associated viruses – replace with honey bee-associated viruses.
Line 74. If prewinter colony losses are related to winter losses - this statement sounds kind of strange because there is no way prewinter colony losses to be unrelated to winter losses, considering that no brood is produced in winter.
Line 78-79. loss winter colony losses – replace with overwintering honey bee losses.
Line 82-83. The honey flow variables were regressed against estimates of V. destructor incidences monitored in an independent study. When the authors evaluate the infestation rate with ectoparasitic mite – Spring, Autumn??? The authors taken into account Varroa as the main factor related to winter losses, which is not quite correct, as there are other factors influencing winter losses – honey reserves, pollen load etc.
Line 219. Table 1 – ns, but this abbreviation is given below the table as not applicable.
Line 287-292. I find it difficult to accept this hypothesis as there are quite a few ambiguities in the statements made.
Author Response
Dear Editor,
The authors of this manuscript evaluated the influence of foraging activity on yearly fluctuations in honey bee losses in Germany.
I have some suggestions to improve the paper.
Line 28. high cumulative honey flow – please, give some details of this statement.
We now define “cumulative” in the description of TrachtNet in MM (which has been rewritten) but we have kept the phrase as it is in the abstract. Cumulative is the sum of net foraging.
Line 30-31. And winter losses were inversely related to autumn losses in the following year - I see no logic in this statement. Please, explain in more detals.
we have amended the text by mentioning that the relationship between loss rates in autumn and winter is positive (we didn’t do that before), hence defining the meaning of “inverse relationship” between winter and autumn of the following year: “Autumn colony loss rates were positively and significantly correlated with winter colony losses, while winter loss rates were inversely related to loss rates in autumn of the following year.”
Line 69-70. associated viruses – replace with honey bee-associated viruses.
done
Line 74. If prewinter colony losses are related to winter losses - this statement sounds kind of strange because there is no way prewinter colony losses to be unrelated to winter losses, considering that no brood is produced in winter.
The paragraph has been completely rephrased (and was indeed confusing). Concerning the statement above, we now write: If pre-winter loss rates predict winter loss rates and if factors influencing these losses can be traced in the foregoing months in a similar manner as winter losses to the preceding summer [20-22] …..
Line 78-79. loss winter colony losses – replace with overwintering honey bee losses.
The paragraph has been rephrased. We replaced “losses” with “loss rates” when appropriate. Or study addresses loss rates (estimated from number of losses). In the previous submission, we did not always consider the difference.
Line 82-83. The honey flow variables were regressed against estimates of V. destructor incidences monitored in an independent study. When the authors evaluate the infestation rate with ectoparasitic mite – Spring, Autumn??? The authors taken into account Varroa as the main factor related to winter losses, which is not quite correct, as there are other factors influencing winter losses – honey reserves, pollen load etc.
The paragraph has been rewritten. Infestation levels with Varroa were estimated by the German bee monitoring programme (not by us). We used the published data together with our hive scale data.
Line 219. Table 1 – ns, but this abbreviation is given below the table as not applicable.
Amended. “-“ means “not applicable”. “ns” (not significant) was missing, and has been added
Line 287-292. I find it difficult to accept this hypothesis as there are quite a few ambiguities in the statements made.
The hypothesis is a thought experiment that is open for discussion and rebuttal. Our study made us think about that to understand colony losses, we likely need to look beyond beekeeping (and agricultural) related mortality factors and also consider bees as natural populations in an ecological context.
We now write: “As a thought experiment, susceptible colonies might be “purged” in winters following years with honey flow conditions favouring high loss rates. This will leave “strong” colonies surviving the winter and lead to lower general loss rates in the subsequent autumn and winter, though these rates will be influenced by honey flow conditions during that year.”
Round 2
Reviewer 3 Report
Please make the previous comments

Author Response
In Figs. 2 and 4: delete the words blue and orange from fig. capture and select show legend from chart elements.
Done. We have also changed the text in fig 2: Associations between autumn loss rates and loss rates of the previous and subsequent winter in Germany, ….
In Fig. 6: delete the words blue and red from fig. capture and select show legend from chart elements.
done
The author can use the following reference throughout the Introduction and discussion section since this article has been compared between two races in harsh condition for foraging activity, stored pollen, worker and drone sealed brood, population size, and honey yield:
Taha, E-K.A.; Al-Kahtani, S.N. (2019). Comparison of the activity and productivity of Carniolan (Apis mellifera carnica Pollmann) and Yemeni (Apis mellifera jemenitica Ruttner) subspecies under environmental conditions of the Al-Ahsa oasis of eastern Saudi Arabia. Saudi Journal of Biological Science, 26(4): 681-687
We have included the reference twice in the discussion (ref:34). As we wrote in our previous reply, we considered the paper in an earlier draft but left it out. We left it out because we considered the paper to deal principally with genetic (subspecies) interactions with the environment rather than with overwintering losses. Due to this consideration, we have not included it as a reference in the Introduction. We have included it in the discussion as a reference to how abiotic variables other than temperature are related to flight/honey flow activity; here we should have included it before.